# Interfacial Engineering Methods in Thermoplastic Composites: An Overview

**DOI:** 10.3390/polym15020415

**Published:** 2023-01-12

**Authors:** Kailashbalan Periasamy, Everson Kandare, Raj Das, Maryam Darouie, Akbar A. Khatibi

**Affiliations:** School of Engineering, RMIT University, Melbourne, VIC 3001, Australia

**Keywords:** interfacial bonding, thermoplastic composites, nanoparticle inclusion, fibre surface treatment

## Abstract

The paper critically analyzed different interfacial enhancing methods used in thermoplastic composites. Although the absence of cross-linked polymer chains and chemical bonds on solidification enables the thermoplastics to be remelted, it creates weak interfacial adhesion between fibre reinforcements and the thermoplastic matrix. The weak fibre-matrix interface bonding reduces the efficiency with which the applied load can be transferred between these composite constituents, causing the composite to fail prematurely. Their need for high-temperature processing, poor compatibility with other polymer matrices, and relatively high viscosity render thermoplastics challenging when used to manufacture composite laminates. Therefore, various methods, including nanoparticles, changing the polarity of the fibre surface by plasma etching, chemical treatment with ozone, or an oxidative attack at the fibre surface, have been applied to improve the fibre/matrix bonding in thermoplastic composites. The fabrication steps followed in these techniques, their progress in research, and the associated toughening mechanisms are comprehensively discussed in this paper. The effect of different fibre-matrix interfacial enhancement methods on the mechanical properties of thermoplastic composites is also deliberated.

## 1. Introduction

Thermoplastic polymers offer a more sustainable solution for fabricating high-toughness components since they can be thermoformed, manufactured at high production rates, and recycled without affecting their physical properties [1]. High-performance thermoplastics, such as polyamide6 (PA6), polyphenylene sulfide (PPS), and poly (ether-ether ketone) (PEEK), are widely used in aircraft components [2] and consumer products [3] to meet elevated temperature service requirements. These polymers, after the addition of fibre reinforcement, exhibit excellent thermo-mechanical properties [4]. Although the fabrication of thermoplastic composites is much more complex and expensive than traditional thermoset composite manufacturing, the introduction of additive manufacturing for thermoplastic composites has created a new image of ease of processing and sustainability. There is a growing trend toward the additive manufacturing of thermoplastic composites for primary structures. Therefore, enhancing the mechanical performance of thermoplastic composites and overcoming manufacturing deficiencies is a primary need for the full utilization of these composites.

Because of long and inert macromolecular chains in thermoplastics, further enhancement of the mechanical properties using copolymerization or blending with other high-performance polymers is minimal [5]. One possible solution for improving the mechanical properties of thermoplastic composites can be achieved by cultivating fibre-matrix interfacial bonding. Fibre-matrix interfacial properties play a vital role in structural integrity and stress transfer in fibre-reinforced composites. Three main interactions govern the adhesive strength at the interface: (i) physical adhesion related to surface energies of the fibre and the matrix, (ii) chemical bonding, and (iii) mechanical interlocking created by rough fibre surfaces. When the surface energy of fibre is higher than that of the polymer, good wetting and an increased contact surface between the fibre and matrix can be achieved. Research shows that surface energies also affect the stability of fibre-matrix bonding after composites consolidation [6]. Due to the lack of covalent bonding and high melt viscosity of thermoplastic polymers, enhancing their interfacial properties is primarily limited to physical interactions.

Comparatively, more research has been conducted on the interfacial enhancement of thermoset composites than on thermoplastic counterparts. A comprehensive literature survey shows that methods used to enhance interfacial properties in thermoplastic composites can be broadly classified under two streams: nanoparticle inclusion and fibre surface treatment. Nanoparticles can be deposited on fibres through various techniques, such as electrophoretic deposition (EPD), chemical vapor deposition (CVD), spray gun process, nanoparticle grafting, and flame synthesis. On the other hand, fibre surface treatment can be conducted using wet-chemical or dry-mechanical interlocking methods. While the wet-chemical technique comprises a chemical coating, chemical grafting, acid oxidation, polymer sizing, and anodic oxidation, the dry-mechanical interlocking can be achieved through plasma, ozone, gamma irradiation, and heat treatment processes. The synergistic effects of combining the fibre surface treatment and the nanoparticle addition have also been reported in the literature.

Additive manufacturing of thermoplastic composites offers new solutions to overcome the limitations of traditional composite fabrication techniques. However, reducing the microscopic voids and enhancing bonding between filament interlayers in additively manufactured composites are among the challenges [7]. The mechanical performance of the polymer filament used in 3D printing has been improved vastly by incorporating continuous fibre-reinforced polymer filaments. The specific strength and stiffness of continuous fibres increased the mechanical performance of the 3D-printed parts. However, the critical problem in 3D-printed polymers or fibre-reinforced polymer filaments is the formation of microscopic voids within the filament and between the individual filaments during the additive manufacturing process [8]. Applying interfacial engineering techniques like fibre treatment or nanofiller inclusion method during the manufacturing of polymer filaments could support the development of 3D-printed parts with reduced voids and improved interfacial and interlayer bonding. 

As presented in Figure 1, the literature survey shows that the study of nanofiller integration and fibre surface treatment in thermoplastic composites has increased in recent years. Even though interfacial engineering in thermoplastic composites began with fibre surface modification using rare earth solution treatment in 2002 [9], the relative growth in fibre treatment methods is considerably slower compared to the integration of nanofillers. Among the different fibre surface treatment methods applied to thermoplastic composites, polymer sizing is the most-used method, followed by gamma irradiation, electron beam irradiation, plasma treatment, chemical grafting, and coating methods. Only a few research works were found that used ozone treatment on fibres. Heat treatment on fibres is a recently-developed method that uses a conventional oven, IR heaters, or a naked flame to treat the fibre surface and increase its ability to bond with the thermoplastic matrix. The heat treatment of fibres has been shown to improve fibre surface roughness, wettability, and interfacial properties. The use of plasma treatment has been gradually declining over the past decade because dry fibre treatment methods like using plasma are known to damage the fibre material. Thus, the number of publications involving the wet chemical method is relatively higher than the dry mechanical interlocking method. 

According to this literature survey, the first nanofiller addition was conducted in 2010 for the fabrication of a multi-walled carbon nanotube (MWCNT)/polyimide (PI) composite film using the electrophoretic deposition (EPD) process [9]. Direct immersion sizing is the most-used technique in interfacial engineering involving nanofillers. Recently developed methods using spray guns and flame synthesis exhibit the ability to add nanomaterials with ease and fewer chemicals. These new techniques also result in higher interfacial and interlaminar properties than other methods, such as electrophoretic deposition (EPD) and chemical vapor deposition (CVD). 

To fully utilize all potential benefits of thermoplastic composites and overcome manufacturing deficiencies of 3D-printed composites, it is necessary to tailor the fibre-matrix interfacial properties, which requires critical analysis of available fibre-matrix interfacial modification techniques. This review primarily focuses on analyzing engineering methods to enhance fibre-matrix interfacial bonding in thermoplastic composites made using high-performance continuous fibres. Available interfacial engineering techniques for these composites are critically analyzed, and their associated mechanisms (i.e., interdiffusion, chemical bonding, and mechanical interlocking) have been identified. This review aims to present a concise summary of interfacial enhancing techniques applied to thermoplastic composites and provide details on recently developed techniques yet to be appropriately tested. The review can help bridge the research gaps and direct future research efforts in the most productive direction.

## 2. Nanofiller Inclusion Methods

The nanoparticles coated over the fibre affect its surface roughness and improve wettability and resin impregnation through mechanical interlocking between fibre and matrix [10]. The commonly used nanoparticles include single- and multi-walled carbon nanotubes, graphene oxide, thermally reduced graphene oxide, and silver nanowires. Titanium dioxide, tungsten disulfide, silica, and rubber nanoparticles have also been extensively used. Adding these nano fillers to the fibre surface reduces the crack propagation in the fibre-matrix interfacial region via toughening mechanisms like crack front pinning, fibre crack bridging, particle bridging, particle/fibre joint fracture, nanoparticle pull out, crack deflection, crack meandering, and micro crack toughening [11]. Available techniques for surface coating of continuous fibres using various nanoparticles include electrophoretic deposition (EPD), chemical vapor deposition (CVD), direct immersion, spraying, nanoparticle grafting, and flame synthesis. These techniques, together with their advantages and limitations, are discussed in the following sections.

### 2.1. Electrophoretic Deposition

In the electrophoretic deposition (EPD) method, charged micro/nanoparticles (mixed in the electrolyte suspension) move towards the carbon fibre with an opposite charge under the application of electric current. The charged nanoparticles are then physically deposited on the fibre surface in a random orientation [12]. 

Liu et al. [10] increased the interfacial bonding between carbon fibre (CF) reinforcement material and polycarbonate (PC) thermoplastic matrix by depositing Multiwalled carbon nanotubes (MWCNTs) over the carbon fibre surface using the ultrasonic-assisted EPD method. The carbon fibres were desized by heating at 600 °C in a nitrogen gas atmosphere. The desized carbon fibres were then passed through the pulse ultrasonic-assisted electrophoretic depositor. It had a mixture of ultrasonically diluted acid oxidized CNTs, isopropanol and ammonium bicarbonate (NH_4_HCO_3_). The ultrasonicator was used to create a uniform dispersion of CNTs in the electrolyte unit, precluding CNT agglomeration during its deposition on the fibre surface. The CNT-coated carbon fibres were then immersed into a waterborne emulsion, a mixture of 50% bisphenol A diglycidyl ether epoxy resin and 50% polycarbonate. The final CNT/carbon hybrid fibres were oven-dried and then irradiated by Q-switched neodymium-doped yttrium aluminium garnet (QS Nd: YAG) laser to heat the binder polymer on the carbon fibres. The laser irradiation effectively removed the extra polymer binder from CNT/CF hybrid fibres. The CNT length increased the resin infusion and the mechanical interlocking between fibre and matrix. Among the different CNT lengths tested, CNTs ranging from 1–5 µm were found to overlap and entangle to form a uniform network for resin infiltration with a CNT suspension concentration of (0.5–1) wt.%. Using the process mentioned above, the interfacial shear strength (IFSS) of the MWCNT-modified sample increased by 68% more than the bare CF-reinforced sample. The integration of MWCNTs at the interface region dissipated the energy of the propagating crack by pinning the crack and deflecting the crack tip.

In another similar research project, short carbon fibres coated with MWCNTs using EPD were used as reinforcements in a PPS matrix composite [13]. With optimized EPD parameters (30 V and 5-min deposition time), the coated fibres were prepared as shown in Figure 2, and then hybrid fibres were melt-blended with PPS in a twin-screw extruder. They were pelletized, compression molded at 320 °C and cut into the required dimensions for property testing. The results showed improved IFSS and electrical conductivity by 42% and 78%, respectively. 

To reduce the peeling of CNTs from the CF surface after the EPD process, CNT-deposited CFs were immersed in a pre-impregnation solution made of PC/dichloromethane (CH_2_Cl_2_) [14]. These treated CFs were hot-pressed with PC films to prepare CF-PC laminates. This EPD process, combined with pre-impregnation treatment, improved the contact area and wettability of the CF surface, which in turn improved the interface connection between the CF and PC matrix. This technique increased the tensile strength, tensile modulus, impact strength, storage modulus, and electrical conductivity by approximately 47%, 58%, 269%, 78%, and 54%, respectively.

In another work, EPD and poly (phthalazinone ether ketone) (PPEK) sizing was combined to deposit and attach the CNTs on CFs more firmly [15]. This CF-CNT was used to reinforce the PPEK matrix and tested for IFSS. EPD, followed by thermoplastic sizing on CF, increased its surface wettability, thereby increasing the IFSS by 36%.

Wu et al. [9] fabricated MWCNT/polyimide (PI) composite films through electrophoretic deposition of carboxylated-MWCNTs and poly (pyromellitic dianhydride-co-4,40-oxydianiline) (PMDA-ODA). These negatively charged MWCNTs, and PMDA-ODA were deposited on a positive anode electrode under an electric field and formed MWCNT/PI composite film. The presence of MWCNTs in the PI matrix decreased the surface resistivity by approximately 100% and increased the tensile strength and modulus by 14% and 50%, respectively. 


*Limitations:*
When acid-oxidized graphene nanoplatelets (GNPs) were deposited on CFs, this led to an increase in the interfacial strength between CF and epoxy. The improvement in the interfacial strength was attributed to the creation of a multiple crack propagation zone [16]. It should be noted that the deflection of cracks into various paths is more prevalent in thermosets than in cases of thermoplastic matrices due to the relatively higher intrinsic toughness of thermoplastic matrices. The stiffness and brittle nature of the thermoset matrix support the nanoparticles to deflect the propagating crack in multiple pathways, which is rare in the case of relatively ductile thermoplastics.The EPD is an automated process suitable for large-scale industrial applications, and it has the potential to damage fibres.The EPD technique may not achieve a uniform deposition of the nanoparticles leading to agglomeration.The EPD method is suitable for fibre reinforcements with relatively high electrical conductivity, such as carbon fibres. This technique is unsuitable for electrically insulative fibres such as glass and aramid [12].


### 2.2. Chemical Vapor Deposition (CVD)

In the chemical vapor deposition (CVD) technique, the fibre is placed inside a quartz tube with a volatile catalyst precursor and heated using a hydrocarbon source like ethane or methane gas [17]. The reaction burns the precursor and allows it to settle after vaporization on fibre material, thereby depositing CNT onto fibre surfaces, as shown in Figure 3. Sometimes, carrier gases like hydrogen and argon are introduced along with hydrocarbon sources to improve the reaction rate in the quartz tube, as shown in Figure 3.

Jianguo et al. [17] deposited CNTs on discontinuous and randomly distributed CFs using the chemical vapor deposition process before mixing them into a polyimide matrix to form composite laminates. The CFs were nitric acid-treated to create surface functional groups before carrying out the CVD process. Benzene was used as the hydrocarbon source, and the CVD process was conducted at 700 °C for 45 min. The tensile strength and impact strength in the modified composites improved by 30% and 29%, respectively, for the CNT weight concentration of 3 wt.%. However, the tensile and impact properties decreased with an increase in nanofiller inclusion beyond 3 wt.%. The SEM images of fracture morphology showed that the carbon fibres pulled out from the matrix in the baseline laminate, leading to annular cracks in the matrix, which led to delamination. However, when the CNTs were added to the CF/PI composite, a brittle fracture with a flatter fracture surface was observed. The pull-out length of CF from the matrix was decreased by the mechanical interlocking between fibre and matrix formed by CNTs. It produced fewer annular matrix cracks with shorter CF pull-out from the matrix.

In another CVD method, CNTs were deposited on CFs using Fe (C_2_H_5_)_2_ (ferrocene) as a catalyst and ethanol as a carbon source [19]. The process was conducted at 750 °C for 900 s. The CVD process created a continuous and dense deposition of CNTs on CF as shown in Figure 4. Catalytic deposition in the CVD process created defects in CFs, reducing the CNT grafted fibre strength by 12%. PP (polypropylene) matrix penetrated the CNTs, which increased the IFSS by 35% for modified CF/PP samples. The fibre/CNT joint fracture was observed to be the dominant failure mode for the grafted fibres.

Tanaka et al. coated nickel catalyst particles on CF through the electrolytic nickel-plating method. These catalyst-coated CFs were used with ethanol as a carbon source and argon as the carrier gas in the CVD process to deposit CNTs on the CF surface. The CNT grafting on unsized CF increased the mechanical properties of CF/PA6, such as IFSS [20] and Charpy impact value [21], by 50% and 10%, respectively. It was also noted that the presence of CNTs in the interfacial region reduced the IFSS degradation of CF/PA6 samples when they were subjected to water absorption [22] and high-temperature tests [22,23]. The above results proved the significant role of CNTs in increasing the mechanical interlocking between CF and PA6 matrix.


*Limitations:*
The CVD method is very effective in forming an even deposition throughout the fibre surface, but it can result in thermal-induced fibre damage, ultimately degrading the composite mechanical properties [18].While thermosetting polymers can form covalent chemical bonds with the treated nanoparticles or with the surface functional groups on the carbon fibre surface, thermoplastic polymers can form only non-covalent chemical bonding [24]. The absence of strong chemical bond formation could be the primary reason behind the relatively low research effort to enhance interfacial bonding in thermoplastic composites using CVD and other chemical-based techniques like chemical coating or nano grafting.While the CVD method provides a homogeneous deposition at the nanoscale level, the requirement for hydrocarbon gases and quartz tube testing apparatus makes it an expensive approach.Catalyst contamination during processing can also be a drawback in the CVD process [25].


### 2.3. Direct Immersion Sizing (DIZ) of Nanoparticles

Direct immersion sizing is the most applied method to increase the adhesive strength between fibre and thermoplastic matrix in composite laminates. It involves directly dipping fibre material into the sizing unit containing nanoparticles mixed with a polymer solution [5] or other supporting chemical agents [26]. A thorough literature survey shows that the nanoparticles most used with DIZ are graphene oxide (GO), multiwalled carbon nanotubes (MWCNT), and nano-silica particles.

#### 2.3.1. GO Nanoparticles

Chen et al. [26] enhanced the interfacial bonding between CF and PEEK using a mixture of polyetherimide (PEI) and graphene oxide sizing on CFs in different mixing ratios. The bare carbon fibres were pulled through the sizing agent, a mixture of PEI granules dissolved in N-methyl-pyrrolidone (NMP) and GO. The carbon fibres coated with PEI and GO were then IR heated at about 300 °C. After drying, the sized CFs were pulled through a suspension mixture of PEEK, Triton X-100, and deionized water, as shown in Figure 5. After PEEK suspension, carbon fibres were placed in an oven and consolidated to produce CF/PEEK prepregs. The CF/PEEK prepregs coated with PEI and GO were stacked in unidirectional orientation and hot-pressed to produce composite laminates.

As shown in Figure 6, the presence of PEI on the CF surface slightly reduced debonding between fibre and matrix. However, adding GO nanoparticulate to the PEI sizing further improved the fibre wettability and fibre/matrix adhesive quality. The interfacial shear strength (IFSS) increased by 14% for CF/PEEK composite laminate containing CFs coated with PEI only. However, when GO sheets were added together with PEI sizing on CFs, the IFSS of the composite laminate increased by 44%. The optimal amount of GO loading for the IFSS was found to be 10 wt.% of the PEI sizing agent. Extensive fibre pull-out and interfacial debonding were the predominant failure mechanisms for control samples. These failure mechanisms were less prominent after incorporating GO at the CF/polymer matrix interface. In another set of research, the interfacial properties of carbon fibre/polypropylene (CF/PP) composites were improved by coating graphene oxide and branched-polyethyleneimine on the CF surface via a layer-by-layer assembly process [24]. The nanomaterial deposition and thermoplastic sizing on CF increased the IFSS, tensile strength, storage modulus, and elongation at the break of CF/PP composite by approximately 102%, 48%, 20%, and 101%, respectively.

Treating nanoparticles in acid can create physical and chemical modifications on the nanofiller surface, enhancing its interaction with fibre and matrix. However, the functionalization of nanoparticles cannot form a chemical bonding with a thermoplastic matrix unless they have been functionalized using a compatible polymer or a chemical agent. In work done by Liu et al. [27], a novel sizing compound containing polyether sulfone (PES), functionalized GO, and dimethylacetamide (DMAC) was integrated to increase the interfacial strength of CF/PES composites. The GO nanoparticles were covalently functionalized by 4,4’-diamino diphenyl sulfone (DDS) and 4,4’-diamino diphenyl ether (DDE), which have similar chemical structures to the PES matrix. The PES/GO-DDS sizing increased the IFSS by 74% and ILSS by 40% compared to the baseline specimens.

A biodegradable polymer called shellac flakes was dissolved in iso propanol, and plasma-treated CFs were immersed in that shellac solution. These modified CFs were then annealed at 700 °C in a tube furnace under vacuum conditions to form shellac-derived reduced graphene oxide (SRGO) on the CF surface. These SRGO-grafted CFs were then coated with a sizing agent made of PA and SRGO. The physical and chemical bonding mechanism that occurred at the interphase due to the interaction between the sizing agent and SRGO increased the ILSS, flexural strength, and modulus of CF/PA6 composite by 71%, 73%, and 84%, respectively [28].

Similarly, sulfonated poly (ether sulfone) (s-PSF) was mixed with GO and used as a sizing agent to increase the interfacial bonding between CF and PEEK. Carboxyl groups on GO formed hydrogen bonds with oxygen-containing functional groups of s-PSF, creating a well-dispersed sizing solution. This sizing agent also withstood the high-temperature processing of CF/PEEK laminate fabrication. The results showed that the ILSS, flexural strength, and modulus improved by 128%, 102%, and 83%, respectively [29].

Desized CF was cut into 3–5 mm and added to the GO aqueous suspension to prepare GO-coated short carbon fibres (GO-SCF). GO-modified PP was obtained by extrusion and melt mixing of GO and PP. Then, the GO-SCF-reinforced GO-PP composite was prepared using the extrusion and injection molding process. The chemical reaction and mechanical interlocking between GO on the SCF surface and GO in the PP matrix increased the flexural, impact, and tensile strength by approximately 33%, 31%, and 94%, respectively [30].

SCFs desized using acetone were treated using thermoplastic polyimide (TPI), graphene oxide (GO), and GO-TPI hybrid sizing to obtain the TPI-sized SCF, GO-sized SCFCF, and GO-TPI-sized SCF, respectively. These three types of sized SCFs were mixed individually with PEI using extrusion and injection molding to prepare SCF-reinforced PEI composite. Among the TPI, GO, and TPI-GO hybrid sizing on SCF, the TPI-GO hybrid-sized SCF/PEI sample exhibited a better tensile strength of about 13% increment to the control sample SCF/PEI [31]. It is due to the crack-blocking effect of GO and the miscibility of TPI with PEI.

Large-size reduced graphene oxide (LRGO) was mixed with ammonium polyphosphate (APP) to produce LRGO-sized APP (LRAPP). These LRAPPs were blended with thermoplastic polyurethane (TPU) using a roll mixing mill and hot-pressed to form thin sheets. Compared to APP-modified TPU, the addition of LRAPP in TPU increased the tensile strength of the polymer up to 190% [32]. LRGO acted as a linking agent in the interfacial region between APP and TPU.

In another study, GO modified with ferrites (GO-Fe_3_O_4_) was mixed with acrylic-styrene (AS) sizing emulsion and coated on CFs [33]. The sized carbon fibres were introduced to the magnetic field to orientate and modify the morphology of GO-Fe_3_O_4_ sizing. These sized- and magnetic-field-treated CFs were then used to reinforce the PP matrix using a hot compression molding process. The presence of oriented GO-Fe_3_O_4_ in the AS sizing agent increased the ILSS of CF/PP composites by 32% [33].

#### 2.3.2. MWCNT Nanoparticles

A methyl methacrylate (MMA)-based liquid thermoplastic resin was used by Shanmugam et al. [34] to fabricate ultra-high-molecular-weight polyethylene (UHMWPE)-reinforced Elium composite laminate. Benzoyl peroxide (2 wt.%) was used as a polymerization initiator for Elium 188 resin. A mixture of deionized (DI) water, polydopamine (PDA), hydrochloric acid, and 0.03 wt.% of MWCNTs were used as a sizing solution. Transverse fibre bundle test and mode I fracture toughness test were conducted to determine the effect of sizing on fibre-matrix interfacial bonding [35]. The fracture toughness improved by 42.5% after sizing due to stick-slip behavior and crack-blunting. The toughness enhancement can be attributed to the strong fibre-matrix adhesion enhanced by MWCNTs. In another study, glass fibres (GFs) were sized using a mixture of surfactant Triton X100, DI water, carboxylic functionalized MWCNTs, and polyphenylene sulfide (PPS) [36]. The one wt.% of MWCNTs used for sizing increased the thermoplastic composite’s fracture toughness and interlaminar shear strength by 25% and 23%, respectively.

The carbon nanotube sizing on fibre material was used as a healing sensor along with improving the fibre-matrix bonding [37]. MWCNTs were coated on GFs using the immersion sizing method, and the sized GFs were then used to reinforce the ethylene-vinyl acetate (EVA) thermoplastic matrix. The MWCNTs in the interfacial region increased the IFSS by 49% and acted as a sensor to monitor and heal the cracks developed in the interfacial region. The interfacial damage was healed by heating the MWCNTs using electric power [37].

Wu et al. [38] used the sizing process to introduce silanized CNTs on the CF surface and increase its bonding with methyl phenyl silicone resin (MPSR). Silane functionalization on CNTs increased its dispersion in the sizing agent and improved the interfacial adhesion of CNTs with CF and MPSR. The test results of modified samples showed an increase of 47% in ILSS and 31% in impact toughness.

A sizing agent made of PI and loosely packed CNT arrays was used to modify the CF surface to increase the wettability and polarity of the CF surface [39]. The CF/PEEK laminate prepared from the sized fibre exhibited a significant increase of 71% in ILSS, 63% in flexural strength, and 70% in flexural modulus. The loosely packed CNT arrays were proved to be better than the dense CNT networks in improving the flexural properties due to the more effortless flow of viscous PEEK melt into the loosely-packed CNT arrays.

Silanized GFs were immersed in a sizing unit containing oxidized MWCNTs dispersed in it, and then these sized GFs were used to reinforce the PPS matrix. The chemical bonding was formed between amine groups on the GF surface, and carboxyl groups linked with MWCNTs created strong MWCNT anchor points on the GF surface, increasing the load transfer between fibre and matrix. The results showed that the tensile and flexural strength increased by 126% and 155%, respectively [40].

GFs were added to the PEI-CNT dispersion before being used to reinforce the PA6 matrix. The GF/PA6 composite was prepared by a twin-screw extruder and injection molding process [41]. The network structure and high surface reactivity provided by PEI-CNT sizing on the GF surface increased the GF-PA6 interfacial performance, thereby enhancing both the tensile strength and flexural strength of modified composites by 7%.

#### 2.3.3. Nano Silica Particles

Chen et al. [5] prepared a complex PEEK suspension made of fine PEEK powders, nano-silica materials, emulsifier Triton-100, and deionized water. The mixture was stirred and ultrasonically vibrated. The CFs were immersed in this complex PEEK suspension and consolidated using an oven to form CF/PEEK prepregs. These prepregs were stacked and hot-compressed to form a composite laminate. The nano-silica added to the CF/PEEK composite laminate acted as a strong obstacle to crack initiation and propagation. The load transfer in the interlaminar region was also enhanced during loading. Nanomodification increased the interlaminar shear strength, flexural strength, and flexural modulus by approximately 16%, 14%, and 9%, respectively. Similarly, silica nanoparticles improved the fibre/matrix interfacial bonding five times in GF/PP thermoplastic composites after nanoparticle sizing [42].

Nano-silica sizing was also found to promote hydrophobic fibre surface properties. When the nano-silica-sized GFs were used in GF/polymerized poly (cyclic butylene terephthalate) composites, it was found that after hydrothermal aging, the water absorption and effect of heat and moisture on the properties of the composite laminates were diminished [43]. Thus, the silica nanoparticles decreased the strength degradation of the composite laminate due to the hydrothermal effects [44].

#### 2.3.4. Mixing of Nanoparticles

Sharma et al. [45] studied the synergistic bridging effects of mixing GO and CNT with polycarbonate (PC) matrix resin. The nanofiller-modified resin was applied over aramid fibres (Kevlar 49) and compression molded. The resulting specimens showed improved static and dynamic mechanical properties. Similarly, a polyimide (PI) suspension containing Fe_3_O_4_ nanoparticles and reduced graphene oxide (RGO) was prepared [46] and used as a sizing agent for the CFs. The sized CF reinforcements were combined with PI to form the CF/PI composite laminate. Fe_3_O_4_ nanoparticles prevented the agglomeration of graphene and increased the ILSS by 159%.

#### 2.3.5. Innovative Non-Conventional Sizing Approaches

Some unique and unconventional interfacial engineering approaches have also been introduced under the Direct Immersion Sizing (DIZ) category, including:Immersing carbon fibres in hydroxylated PEEK grafted multiwalled carbon nanotubes (HPEEK-g-MWCNT) solution to improve the bonding strength with the PEEK matrix [47]. Flexural strength and ILSS were increased by 94% and 55%, respectively, for the modified samples. The formation of chemical bonding between the PEEK matrix and HPEEK-g-MWCNT, along with the mechanical interlocking between fibre and matrix provided by MWCNT, contributed to the improvement in interfacial bonding between CF and PEEK.Creating a sizing agent made of polyetherimide (PEI) and in situ-grown nanocrystals made of zeolitic imidazolate framework-67 (ZIF-67) to enhance the strength of CF/PEEK composite laminates [48]. Miscibility of PEI with PEEK and rough structure of ZIF-67 helped to increase the IFSS between CF and PEEK up to 41%.Using the pie-shaped PEI nanoparticles to size the CF surface and heating the PEI-coated fibres to melt a few nanoparticles to increase the surface energy of CF when used to fabricate thermoplastic composites [49]. PEI surface modification increased the IFSS for CF/PVC, CF/PC, CF/PA6, CF/PP, CF/PA66, and CF/PEI by 21%, 38%, 53%, 50%, 43%, and 58%, respectively. The compatibility of thermoplastic resins and PEI coating increased the IFSS between CF and thermoplastic resins. The effect is even better if the surface of PEI coating is in nanoparticle morphology.Modifying the short flax fibre surfaces using cellulose nanocrystals, xyloglucan CNC/XG, and maleic anhydride-grafted polypropylene (MAPP) coupling agent. This CNC/XG biomass by-product adsorption on the flax fibres and the covalent bond provided by the MAPP coupling agent with the fibres enhanced the bonding strength between flax fibre and polypropylene (PP) in flax fibre/PP bio composite [50]. The work of rupture of the flax fibre/PP samples measured by micromechanical tensile tests was improved by 13% and 22% for CNC and CNC/XG treatments, respectively.


*Limitations:*


One prominent shortcoming of the Direct Immersion Sizing method is the variability of the coated thickness. If the dispersion of nanoparticles in the sizing unit is not adequately regulated, agglomeration and uneven distribution of nanomaterials over fibre surfaces can occur.

### 2.4. Spray Gun Technology

The spray gun technique is used to spray nano-fillers mixed with volatile solvents like ethanol on a fibre surface at a specific spraying speed to size the fibre surface with nanoparticles, as shown in Figure 7 [51]. Although this method has not been used frequently in the research, it proves to be effective in creating strong physical adsorption between fibre and matrix with no chemical bonding between them.

Cortes et al. [52] enhanced the electrical conductivity of carbon fibre-reinforced polyether ketone ketone (PEKK) by adding silver nanowires using two fabrication methods: film stacking and powder impregnation through spray coating. The powder impregnation method produced a more homogenous material with good dispersion of nanowires. This method gave conductivity up to 250 S/m for a 2.5% volume of Ag nanowires. In the case of film stacking, PEKK impregnation was poor since the silver nanowire film only stayed on the fibre surface. Furthermore, due to the high viscosity of PEKK, poor penetration into carbon fibre tows created a heterogeneous distribution. Meanwhile, the powder impregnation using the spray gun resulted in improved impregnation with no voids, as shown in Figure 8. The inclusion of silver nanowires created a percolation mechanism in CF/PEEK laminates which further enhanced the electrical conductivity.

Likewise, CNTs were sprayed on CF/PEEK prepregs to enhance their electrical and thermal conductivity [51]. A 0.5 wt.% of CNTs proved to be the optimal mixing concentration in CF/PEEK laminates. The CNTs generated an effective conductive pathway between fibre and matrix. Flexural strength, flexural modulus, and ILSS were increased by approximately 25%, 24%, and 36%, respectively, after CNT spray deposition. Crack bridging through fibres and nanoparticles was observed as the predominant toughening mechanism.

He et al. [53] sprayed a mixture of nano-silica and poly (ɛ caprolactone) PCL onto basalt fibres to enhance the mode I interlaminar fracture toughness (ILFT) of basalt fibre reinforced poly lactic acid (PLA) composite laminates. The hybrid coating having PCL and nano-silica mixed in a ratio of 1:5 gave the best performance. The crack deflection was the primary toughening mechanism observed. The SiO_2_ and PCL were constructed as rigid and flexible phases in the interfacial region. The nanofiller modification improved the tensile strength and mode I ILFT by 29% and 110%, respectively.


*Limitations:*


Controlling the thickness of the coating and creating uniformly distributed nanoparticles is challenging while using spray gun technology.

### 2.5. Grafting of Nanoparticles

The grafting of nanoparticles takes place by coating the fibre surface with an adhesive coupling agent and bonding the nanoparticles over the adhesive chemical compound. The presence of the nanoparticles, along with the coupling agent at the fibre surface, promotes the formation of chemical bonds between the reinforcements and the polymer matrix [54,55]. In some cases, the fibre reinforcements are chemically or mechanically treated to increase fibre wettability. Improved fibre wettability enhances the adhesion of fibre with the polymer matrix and the surface-deposited nanoparticles via mechanical interlocking and chemical bonding [56].

Kim et al. [54] grafted zinc oxide nanorods (ZnO NR) on woven carbon fibres (WCFs) and applied a CNT-modified PEI silane coating onto the fibre surface. The grafting of nanorods and silane coating enhanced the mechanical strength of WCF/polyamide 6 (PA6) composite laminates. The fibre surface was initially plasma treated to desize, add functional groups, and increase nano-pits, as shown in Figure 9. After accomplishing the ZnO NR growth and silane coating process on the fibre surface, ultra-fast thermoplastic resin transfer molding (RTM) was used to fabricate WCF/PA6 composite laminates. The RTM was done using the caprolactam (CPL) monomer of PA6. Since the melt viscosity of CPL is about 100 times lower than that of thermoset resin, an excellent impregnation between hexagonal wurtzite structured ZnO nanorods was achieved. The ZnO NRs and CNTs maximized the mechanical interlocking effect of the fibre with the PA6 matrix, whereas the amine groups in the PEI silane coating formed a covalent chemical bond with the PA6 matrix. The integration of ZnO NR, CNTs, and silane at the fibre/matrix interface of the multiscale hybrid composite enhanced the fibre/matrix in-plane shear strength, tensile strength, and tensile modulus by approximately 107%, 41%, and 32%, respectively.

In a recent study, graphene oxide (GO) was grafted onto the CF surface by two synthetic routes with hexamethylene diisocyanate (HDI) tri-polymer as the coupling agent [55]. The first method utilized HDI tri-polymer to modify GO. The modified GO was then grafted onto the oxidized CF. The other route utilized an HDI tri-polymer to alter the oxidized CF surface and graft GO onto that modified CF surface. The first route proved more effective in improving the interfacial shear strength between CF and PA6 matrix. The interfacial shear strength increased by approximately 40%. The HDI tri-polymer acted as a coupling agent between CF and GO by connecting the two composite constituents via covalent bonding. The GO contributed to the combined effects of wettability and mechanical locking force. In another work, silica nanoparticles were grafted on the CF surface and used to fabricate CF/PA6 composite material [56]. The oxidized CFs were modified with 3-aminopropyltriethoxysilane and then grafted with silica nanoparticles dispersed in the KH560 coupling agent. The inclusion of SiO_2_ nanoparticles of size 30 nm at the fibre/matrix interface increased the interfacial strength by over 200%, from approximately 13 to 39 MPa.

Due to the high viscosity of the thermoplastic matrix, nanofillers are not readily applicable in these composites, and special treatment is required. In a study to anchor MWCNTs on CF, Cheon et al. [57] used the nitric acid treatment on MWCNTs and flame treatment on CF that improved surface functionalities. Flame treatment was preferred for desizing and adding oxygen-based functional groups on CF surfaces. The research was performed via two approaches: mixing MWCNTs with PA6 matrix using cryogenic pulverization and anchoring MWCNTs on flame-treated CFs with silane coating. The latter approach exhibited improved performance under impact and short beam bending conditions. Using 3 wt.% of MWCNTs increased the impact resistance by 91%; meanwhile, the integration of 1 wt.% of MWCNTs improved the ILSS by approximately 34%.


*Limitations:*


The main concerns for grafting nanoparticles on the surface of continuous fibres are its excessive usage of toxic chemicals and difficulty in processing a large area of fibres. In addition, achieving a well-controlled grafting density, molecular weight, and molecular weight distribution is challenging in this method.

### 2.6. Flame Synthesis of CNTs

In more recent studies, CNTs were grown on fibre materials using flame treatment of the fibres wetted with catalyst solutions. Although CVD is the most utilized process for growing CNTs on continuous fibre surfaces, carbon nanotubes have a poor affinity for polymer matrices due to their inert chemical properties. On the other hand, flame synthesis forms oxygen-rich functional groups over CNT during the growth process. Experimental observations show that the flame synthesis method gives better interlaminar fracture toughness than other methods like the spraying of CNTs, CVD, introducing CNF powder, and CNF paste but not better than through-thickness stitches and Z pinning [58]. As shown in Figure 10, in this method, fibres immersed in nickel chloride are heated using a catalyst flame at a controlled temperature and time to grow CNTs on the carbon fibre surface [58]. The size and density of CNTs can be controlled by varying nickel chloride catalyst molarity and growth time.

Zhao et al. [60] grew CNTs on glass fibres wetted with nickel chloride solution using ethanol flame at 560 °C for 3 min. The same authors investigated the effect of flame synthesized CNTs on the welding of thermoplastic composites [61]. In their work, CFs were wetted with nickel nitrate solution and then heated between 740 °C and 970 °C for 1 min using an ethanol flame. The CNT-grafted carbon fibres and PEI films were hot-pressed to form the CF/PEI composite laminate used as an interlayer during the resistance welding of GF/PEI laminates. The mechanical property of the welded joint was evaluated via tensile testing of the single-lap shear strength (LSS). The LSS value increased by 24% in welded joints incorporating flame synthesized CNTs when compared to the baseline composite. The interfacial shear strength between CF and PEI increased by approximately 47% from 4.8 to 7.0 MPa upon the integration of flame synthesized CNTs. The CNTs increased the wettability of the CF surface by PEI with the help of capillary action.

CNTs were grown on the surface of short glass fibres (SGF) using nickel chloride mixed in ethanol (0.4 mol/L) as catalyst precursor and mixed with the nano grown SGFs with poly lactic acid (PLA) pellets. The mixture was fed into the extruder three times to reduce the filler agglomeration in the PLA matrix. The extruded filament was used to print 3D composite samples through fused deposition modelling (FDM) [62]. Compared to the 3D-printed PLA samples, the tensile strength and Young’s modulus of CNT-modified SGF/PLA 3D-printed samples were increased by 33% and 43%, respectively. The results proved that the flame-synthesized CNTs significantly improved the interfacial adhesion between SGF and PLA matrix in the 3D-printed composite.


*Limitations:*


The flame synthesis of CNTs is a relatively new method that still requires improved understanding and optimization of the process parameters. The distribution, local concentration, and thickness of the deposited CNTs depend on the catalyst, the distance of the substrate from the burner, and the exposure temperature and time, among other variables.

### 2.7. Miscellaneous Methods

In addition to the techniques discussed above, some unconventional methods have been used to engineer the fibre surface using nanoparticles. For example, in a study, 0.1 wt.% of different nanomaterials (i.e., MWCNT, RGO, Graphene, nano clay, and exfoliated graphite nanoplatelet (xGNP)) were individually tested with the plasma-treated CF/PA6 specimen. The nanomaterial was dispersed in ɛ-caprolactam, a monomer of anionic polymerized polyamide 6 (A-PA6). This modified monomer with a catalyst and an activator was passed through CFs using ultra-high-speed resin transfer molding [63]. The results showed that GO surpassed other nanomaterials in increasing the elastic modulus by approximately 29% and xGNP outperformed the remaining nanofillers in increasing the tensile strength, in-plane shear strength, and flexural strength by approximately 18%, 18%, and 29%, respectively

In another study, Rasana et al. [64] melt blended short GFs of length 3 mm and MWCNTs with PP to fabricate PP composites. The melt processing was carried out in two steps. MWCNTs were mixed with molten PP to prepare a masterbatch. The CNT masterbatch, PP, and GF were mixed in a rotating twin-screw extruder in the second step. The extruded strands were pelletized and injection molded into test specimens. The filler modification in PP composites enhanced its tensile strength, Young’s modulus, and elongation at the break by approximately 76%, 127%, and 5.5%. In a similar approach, a twin-screw extruder and injection mold were used to integrate micro fillers (i.e., molybdenum disulfide [MoS_2_], silicon carbide [SiC], alumina) and alumina nanoparticles into short glass fibres (SGF)-, short carbon fibre (SCF)-, and polyamide 66 (PA66)-polytetrafluoroethylene (PTFE) hybrid composite [65]. Adding nano and micro fillers decreased the tensile and flexural strength by approximately 33% and 19%, respectively, due to the increased stress concentration created by the excessive micro and nanofiller inclusion. However, the enlarged surface area of contact between micro and nano fillers increased the impact strength by approximately 18% [66].

Recently, Arao et al. [67] mixed the following nanofillers – silica, CNT, alumina, and nano clay – individually with CF/PP composites via extrusion and injection molding process. It was found that except for nano clay, the remaining nanofillers (i.e., silica, CNT, and alumina) strengthened the interface and thereby improved the tensile properties. Among the nanofillers, 1 wt.% of CNT increased the tensile strength, elastic modulus, and IFSS by approximately 7%, 9% and 186%, respectively. Hwang et al. [68] created a composite powder from PA6 and a nanofiller mixture. The nanofillers were pristine graphene nanoplatelets (P-GNPs) and acid-oxidized GNPs. The PA6/GNP mixture was melt-spun into thin fibres and tested for thermomechanical properties. The results showed that in terms of the tensile properties, PA6 fibres containing acid-oxidized GNPs performed better than those with pristine GNPs. The acid treatment on GNPs increased the surface roughness and surface functionalization of PA6 fibres, which improved the tensile strength and modulus of the composite by 76% and 70%, respectively.

Silica aerogel and rubber-based nanoparticles were also investigated to enhance the mechanical properties of thermoplastic composites. Silica aerogel nanoparticles were mixed with poly (butylene terephthalate) PBT matrix through melt extrusion and injection [69]. It exhibited poor dispersion in the PBT matrix, which decreased the mechanical performance of the PBT matrix. The agglomerated clusters of silica aerogel nanoparticles acted as stress concentration sites and decreased the tensile strength and break elongation by 25% and 58%, respectively [69]. In another study, PEI was used as a non-covalent modifier to modify the surfaces of boron nitride nanoparticles [70]. This non-covalent modified boron nitride (n-CMBN) nanoparticle coated with PEI was mixed with PEEK powder and melt-blended to form a PEEK/n-CMBN polymer nanocomposite. The nanofiller/matrix bonding improved thermal conductivity by four times that of PEEK plastic. The reduction in friction coefficient by 30% after adding n-CMBN to PEEK reflects the better dispersion of nanoparticles in PEEK due to PEI coating on boron nitride nanoparticles [70].

### 2.8. Discussion: Nanoparticle Incorporation

Several key factors affect the fibre-matrix interfacial bonding in a nanoparticle-reinforced thermoplastic composite laminate. The most influential parameters are (i) the wettability of reinforcement material, (ii) the type, size, shape, and quantity of nanofiller added, and (iii) the method through which the nanofiller was deposited onto the fibre surface [71]. The influence of varied nanoparticle integration methods on the interlaminar and interfacial properties of thermoplastic composites were tabulated in Table 1. The advantages and disadvantages were presented in Table 2.

Even though direct immersion sizing is the most commonly used and straightforward process, this method cannot be easily used to mix nanomaterials with high melt viscosity thermoplastic polymers like PEEK. Further, the nanomaterials are not firmly anchored to the fibre surface. The physical adsorption between nanomaterial and fibre surface is more potent in the deposition process like EPD, CVD, and spray gun techniques than the direct immersion sizing deposition method [73]. Non-uniform nanofiller dispersion, and formation of nanoclusters, are recurring drawbacks of direct immersion sizing. The aggregation of nanofillers results in stress concentration, thereby promoting crack initiation and propagation through the interlaminar region [71]. However, new techniques are being developed to reduce agglomeration by adding metallic nanoparticles like Fe_3_O_4_ to the sizing unit [46].

Nanofiller materials like CNTs produce a bridging mechanism between fibre and matrix when deposited on the fibre surface. The resin infusion promoted by the capillary action of nanotubes provides stronger interactions between fibre, matrix, and nanotubes. When these nanotubes of optimal aspect ratio were used to modify the fibre surface, they can create a hierarchical network of nanotubes to increase further interlocking between matrix and fibre [10]. Unlike 1-D CNTs, 2-D nanofillers like GNPs cannot produce entangled networks, but the nanosheets possess high surface energy which supports their adsorption with fibre surface and matrix. However, this higher surface energy of GNPs can cause the agglomeration of nanoplatelets while mixing in the sizing unit [74].

## 3. Fibre Surface Treatment Methods

It is essential to treat the fibre surface while reinforcing polymer matrices because the fibre sizing and surface impurities can affect the interfacial bonding between fibre and matrix [75]. Fibre surface treatment is broadly classified into two types. The wet method involves treating the fibres with chemicals, while the dry process consists of treating the fibres with heat and radiation. The fibre surface roughness is modified in both cases, and active chemical functional groups like hydroxyl, carboxyl, and carbonyl are added. These treatments increase the interfacial bonding between fibre and matrix when the composite laminates are subjected to loading [76].

Before any surface treatment, it is necessary to remove the existing thermoset-based sizing. Li et al. [77] heat-treated carbon fibres at 280, 340, and 370 °C to remove the sizing agents from the carbon fibre surface. These heat-treated carbon fibres were combined with PEEK and prepared for the micro bond test to find the fibre/matrix interfacial shear strength. The sizing agent was eliminated from the carbon fibre surface when heated at 370 °C for 5 min. The preheat treatment of carbon fibres increased the IFSS by approximately 15% while decreasing the content of active carbon atoms on carbon fibre surfaces by approximately 64%. The sizing agents formed weak layers at the interfacial region, weakening the fibre/matrix interfacial adhesion. Desizing is essential in enhancing the fibre/matrix interfacial adhesion in thermoplastic matrix composites. For thermoset matrix, like epoxy resin, the increased concentration of the activated carbon atoms at the carbon fibre surface increases surface chemical activity and interfacial adhesion. However, PEEK does not have functional groups to interact chemically with sizing agents. There is no evidence of chemical bonding between sizing agents and thermoplastic polymers.

In a similar study, Kiss et al. [78] removed an epoxy-based sizing from carbon fibres by infrared-irradiation (IR) in an air atmosphere at 400 °C. The IR-desized CFs were proved to be free from sizing agents by SEM image and thermogravimetric (TGA) analysis. Following the fibre surface treatment, CF/PA6, CF/PPS, and CF/PEEK composite laminates were fabricated using film stacking and hot compression. The fabricated composites were compared against the baseline composite with epoxy-sized fibre reinforcements. Flexural and impact tests showed that thermoplastic matrix composite laminates, prepared from sized CFs, had poor consolidation and wettability, resulting in weak fibre/matrix interfacial bonding. The poor fibre/matrix adhesion in epoxy-sized thermoplastic composite laminate highlights the importance of removing epoxy sizing when fabricating thermoplastic matrix composites.

### 3.1. Wet Chemical Surface Treatment Processes

#### 3.1.1. Polymer Sizing

Polymer sizing is the most common fibre surface treatment process. Polymers of low molecular weight with active functional groups in their molecule are applied onto the fibre surface to improve bonding with the polymer matrix. In the polymer sizing process, the fibre damage is negligible compared to other fibre-treating processes [79]. Selecting polymer sizing to increase the interlocking mechanism with thermoplastic matrices can be challenging. However, compatible sizing agents can produce thermoplastic composite laminates with better interfacial bonding than desized samples [80,81].

Gao et al. [81] used sulfonated PEEK (s-PEEK) as a sizing agent for carbon fibre. Acid-oxidized carbon fibres were immersed in an s-PEEK solution. After evaporating the solvent, the sample was evenly electrostatic sprayed with pure PEEK and heated in an oven to form an s-PEEK-sized CF/PEEK composite laminate. This method increased the interlaminar shear strength (ILSS) and tensile strength by approximately 46% and 11%, respectively. Carbon fibre-reinforced PEEK laminates containing s-PEEK-sized carbon fibres had fewer voids. The optimal mechanical properties were obtained when the weight fraction of s-PEEK was 10%. The 46% improvement in the ILSS is more significant than the results obtained using acid oxidation (29%), plasma modification (16%), and chromate oxidation (26%).

Similarly, the silane coupling agent and s-PEEK were used to modify the GF surface in GF/PEEK composite samples [82]. The surface modification of GF increased the tensile and flexural strength by 21% and 30%, respectively. The amino groups in the coupling agent and the sulfonic acid within the PEEK sizing agent contributed to the increased van der Waals interaction between GF and PEEK matrix.

A recent study used an easily purified semi-aliphatic polyimide (SA-PI) as a sizing agent for CF/PEEK laminates. The SA-PI sizing increased ILSS by 24% due to the improvement in the wettability and surface energy of CFs [83], while the flexural strength of the CF/PEEK composite increased by 8%.

Attempts have been made involving combining different chemicals to create a sizing agent compatible with PEEK to increase the wettability of the CF surface. A soluble PEEK sizing unit was developed using aniline and phenylacetylene [84]. The compatibility of the sizing agent with the matrix and the formation of chemical bonds between the fibre surface and the sizing agent significantly improved the interfacial bonding, increasing the IFSS and ILSS by 138% and 67%, respectively. In another work, the aminated polyether-ether-ketone (PEEK-NH_2_) sizing agent, created through the emulsion/solvent evaporation method, was used to size the CFs. The excellent compatibility of PEEK-NH_2_ fibre sizing with the PEEK matrix enabled the sizing agent to act as a stress transfer bridge at the CF/PEEK interface [85]. It increased the ILSS and flexural strength of the CF/PEEK laminate by 43% and 62%, respectively.

The miscible characteristic of PEEK with PEI makes PEI an essential sizing element for fibre-reinforced PEEK composites. Liu et al. [80] used PEI as a sizing agent for CF/PEEK composite laminates. In one case, the sizing agent PEI was mixed with the NMP solvent, while PEI was mixed with the Triton X100 emulsifier in a separate study. Composites that contain CF handled by emulsion-type sizing agents exhibited better interfacial properties. The IFSS and ILSS increased by 17% and 16%, respectively. Likewise, aqueous PEI sizing stabilized with sodium dodecyl sulphate (SDS) surfactant was used to modify the CF surface in CF/PEEK composite laminates [86]. The polymer sizing on CF decreased its contact angle with deionized water by 9%, indicating increased surface wettability. SEM observations also showed PEI/SDS sizing could create a continuous interface bonding between CF and PEEK matrix.

In some studies, PEKK was also used as a sizing agent to enhance the interfacial interactions between carbon fibre and PEEK [87]. CF was activated using Meldrum’s acid and coated with PEKK to form a hydrogen bond. PEKK also shared good compatibility with PEEK. The above factors contributed to the increase in ILSS, flexural strength, and modulus by 70%, 37%, and 48%, respectively.

Zhang et al. [88] used polyurethane (PU) for carbon fibre sizing before hot pressing with polycarbonate (PC) films to form CF/PC composite laminates. Four types of PU were used to size carbon fibres, namely: blocked polyisocyanate PU, PC-PU, polyester PU and blocked polyether PU. Except for PC-PU, the other PU sizing increased the ILSS by approximately 25%. While the chemical interaction between PU sizing and carbon fibres was attributed to isocyanate’s high reactivity, transesterification promoted the chemical reaction between PU sizing and polycarbonate. In a similar study, Hendlmeier et al. [89] used different sizing agents, such as epoxy, polyamide, and polyurethane, to modify the anodically-oxidized carbon fibres. These modified carbon fibres were used to reinforce an Elium thermoplastic matrix composite. Methyl ethyl ketone peroxide (MEKP) was used as a radical initiator. Samples with polyamide-sized fibres exhibited better interfacial performance than baseline and other-sized samples by increasing the IFSS by approximately 35%.

Yamamoto et al. [90] used the electrophoresis process to deposit poly (methyl methacrylate) (PMMA) particles on the CF surface, enhancing the interfacial adhesion between carbon fibres and thermoplastic resins in the process, as shown in Figure 11. The amount of PMMA particles adsorbed onto the CFs varied based on the electrophoresis duration. The IFSS increased by nearly 64% for CF/PMMA samples with a concentration of 0.152 g/m^2^ PMMA particles [91]. Similarly, the PP colloid, prepared using a surfactant, was deposited onto CFs using electrodeposition [92]. For 0.3 g/m^2^ of PP colloid particle adsorption by CF surface, IFSS increased by approximately 316% and 81% in CF/PP and CF/PA6 samples, respectively.

Dong et al. synthesized carboxylic polyphenylene sulfide (PPS–COOH) with low molecular weight and used this chemical as a sizing agent to improve the interfacial shear strength (IFSS) of CF/PPS composite laminate [93]. The plasma-treated carbon fibre surface was coated with PPS-COOH sizing agent, which introduced the carboxyl functional groups and improved the CF surface roughness. The carboxyl groups in PPS sizing interacted with fibre surface polarity formed by plasma treatment and were compatible with the PPS matrix. This fibre modification increased the tensile strength and IFSS by 15% and 28%, respectively. In another study, the silane-grafted PPS particles were coated on the CF surface using the EPD process to increase the interfacial bonding between CF and PPS matrix in CF/PPS composite laminates. The silane-grafted PPS particles at the interface region created an interpenetrating polymer network and siloxane network to enhance the fibre-matrix interlocking mechanism, which increased the tensile strength and flexural strength by 5% and 17%, respectively [94].

Liu et al. [95] enhanced the interfacial adhesion of CF/PU composite material by coating the CF surface with a hybrid layer of polyethyleneimine/polydopamine (PEI/PDA). The sizing process on fibres was implemented using a one-step dip-coating approach which improved the wettability and surface energy of the fibre surface. It increased their tensile strength, modulus, and toughness by approximately 39%, 59%, and 28%, respectively.

Bio-composites made of polylactic acid (PLA) matrix reinforced with short cellulose fibres were prepared using a twin-screw extruder and injection molding [96]. The cellulose fibres were coated with low molecular weight epoxy-based surface treatment agents. It increased their wettability with the PLA matrix such that the adsorption of the matrix by fibre surface was improved and reduced the hydrolysis and oxidation reaction of the PLA matrix. The storage modulus of the composite sample was increased by 50%.


*Limitations:*


The thin interfacial film, created by wet chemical surface treatment, can generate physical or chemical bonding with the matrix to effectively transfer the load between the fibres and matrix, reducing the effect of the fibre self-defects. Considering thermoplastic composites, selecting a proper sizing agent, and adjusting the processing parameters are the main challenges in engineering the fibre-matrix interfacial properties. In addition, the extra processing time required for washing and drying the fibres, environmental concerns, and health concerns due to the nature of the chemicals are other drawbacks associated with wet chemical methods.

#### 3.1.2. Chemical Grafting and Chemical Coating

In these techniques, fibre surfaces are treated with organic and/or inorganic chemical compounds to initiate a chemical reaction on the fibre surface, which improves the interfacial load transfer property and the bonding strength of fibre with the polymer matrix. Even though this method is relatively complex, involves toxic chemicals, and is costly due to multiple processes, it creates covalent chemical bonds and promotes mechanical interlocking mechanisms [79]. Limited studies in the open literature have explored the effect of chemical grafting and coating on the mechanical properties of thermoplastic composites.

Li et al. [97] grafted epoxy chloropropane (ECP) on the surface of Kevlar fibre to improve the bonding strength of the Kevlar fibre-reinforced polyimide (PI) composite. The ECP grafting modification of the fibre surface was done by soaking Kevlar fibres in the solution of KOH (0.7%) at 30 °C for 2 h, followed by washing and drying. These fibres were then grafted with ECP at 90 °C for 6 h, washed with distilled water, and dried. The friction and wear properties of the polyimide (PI) composites reinforced with surface-treated Kevlar fibre were investigated using a ball-on-block reciprocating tribometer. Experimental results revealed that the ECP grafting reduced the friction and wear of Kevlar/PI composites against the GCr15 steel ball. The ECP grafting decreased the friction coefficient from 0.32 to 0.27 and the specific wear rate from 0.34 × 10^−13^ to 0.26 × 10^−13^ m^2^/N. The worn surfaces of untreated Kevlar/PI showed furrows, whereas the worn surface of treated Kevlar/PI was smooth, indicating improved fibre matrix bonding quality.

Luo et al. [98] prepared a novel composite membrane called PGF membrane by combining poly (vinylidene fluoride) PVDF and glass fibre. They improved the peeling strength of PGF by treating the GF using a silane coupling agent as an initiator and mixing acrylamide monomer (AM) with PVDF. The silane-coated GF was irradiated by a UV lamp under a nitrogen atmosphere for 25 min and coated with a PVDF-AM mixture. The UV irradiation yielded the CH_2_ radical on the GF surface, which reacted with AM in PVDF, forming graft polymerization. This novel composite membrane prepared using an interfacial UV-grafting copolymerization process generated the chemical cross-linking at the composite interface and increased the peeling strength by 33% for an AM concentration of 2 wt.%.

The interfacial properties between CF and poly (arylene sulfide sulfone) (PASS) were improved by grafting amine containing PASS (NH_2_-PASS) chains onto the CF surface [99]. First, CFs were acid treated to form carboxyl groups on the CF surface. Then, the aminated PASS was grafted on the CF surface via an amidation reaction with carboxyl groups on the CF surface. IFSS and ILSS of the modified CF/PASS samples were increased by 28% and 72%, respectively. The results showed that the grafted molecular chains with shorter lengths increased the interfacial bonding through better entanglement with matrix resin.

Unlike the chemical grafting process, which deposits chemical substances over another coupling agent layer on the fibre surface, chemical coating applies chemical agents over the fibre material. It is a relatively more straightforward but less effective process than chemical grafting. Various chemical compounds, including acids and bases, are used in chemical coating to modify the fibre surface and improve its interaction with the polymer matrices [100,101,102].

Ray et al. [101] treated date palm leaf (DPL)-derived fibres using potassium hydroxide (KOH) and corn starch before reinforcing the polyvinyl alcohol (PVA) matrix composite. The composite was fabricated using micro injection molding. The chemical coating on DPL fibres improved the tensile, flexural, and impact strength by approximately 61%, 89%, and 9%, respectively. In a similar study, Panyasart et al. [100] used alkaline and silane treatments on pineapple leaf fibres reinforcing polyamide 6 matrix composites. Young’s modulus and tensile strength increased while the elongation at failure decreased, indicating an increase in composite stiffness. Alkali treatment of fibres produced better tensile results than silane treatment. Alkali treatment of fibres increased the tensile strength and Young’s modulus by 31% and 8%, respectively, to that of the unmodified baseline composite.

#### 3.1.3. Rare Earth Solution Treatment

In rare-earth solution treatment, rare earth elements are either coated on the carbon fibre surface or mixed with the matrix material to increase the fibre/matrix interfacial bonding [102]. Li et al. [103] used a rare earth solution (RES) comprising lanthanum chloride (LaCl_3_) and ethanol at concentrations ranging between 0.1 to 0.5 wt.% to treat the surface of carbon fibres. The CFs were immersed in the RES of PH 5 for 10 min and dried at 120 °C for 4 h. The treated carbon fibre-reinforced polyimide (PI) matrix composite laminate was fabricated using hot molding. The treated composite possesses improved flexural properties compared to the baseline composite. It was also found that the RES treatment of fibres enhanced the flexural properties of composite laminates better than the air oxidation method. At optimal RES concentration of 0.3 wt.%, the flexural strength and flexural modulus increased by 18% and 3.5%, respectively. XPS analysis found that RES treatment increased the oxygen concentration on the CF surface by 174%. RES compounds can react with some functional groups like sulfonic (-SO2-) and carbonyl (C=O) groups on the PI matrix and with hydroxyl (-OH), carboxyl (COOH) and carbonyl (C=O) groups on the CF surface. Hence, the RES surface treatment played an essential role in increasing the surface polarity of carbon fibres and improving their interfacial bonding with the PI matrix.

Cheng et al. [104] treated the CFs using air oxidation and rare earth solution and compared their effect on the mechanical properties of the CF-reinforced polytetrafluoroethylene (CF/PTFE) composite. Air oxidation treatment was done by heating the CFs to 450 °C for 40 min in a box furnace and cooling the fibres down to room temperature in the furnace. RES treatment was done by soaking the CFs in an alcoholic lanthanum chloride (LaCl_3_) for 3 h and drying the treated fibres in an inert atmosphere at 80 °C for 4 h. Treated CFs and PTFE powder were mixed in a three-dimensional mixer and put into the sample mold and compression molded to obtain CF/PTFE composites. The RES-treated CF/PTFE sample showed a 16% improvement in flexural strength. In contrast, the air oxidation-treated CF/PTFE sample showed only a 2% improvement in flexural strength, indicating RES treatment is more effective than air oxidation treatment in increasing the fibre matrix interfacial bonding.

Three types of surface modifiers, namely N-β-amino-ethyl-ɣ-aminopropyltrimethoxysilane coupling agent (SGS), a mixture of silane and rare earth elements (SGS/RES), and rare earth elements surface modifier (RES), were used to treat the GF surface [105]. The treated GFs were mixed with PTFE powders and compression molded to produce GF/PTFE composite samples. RES modifier outperformed the other two surface modifiers in enhancing the tensile property of the GF/PTFE sample by improving the tensile strength by up to 21%.

Similarly, Shi et al. [106] used four types of surface modifiers to treat CFs, namely concentrated nitric acid (conc. HNO_3_), silane coupling agent, rare earth solution (La_2_O_3_), and a mixture of La_2_O_3_ and conc. HNO_3_. The PTFE composite filled with CF treated using (La_2_O_3_ + conc. HNO_3_) mixture exhibited the lowest wear volume loss of 35% compared to the baseline sample. Rare earth treatment made CF difficult to detach during tribological testing, which increased the wear and friction properties of CF/PTFE samples.

### 3.2. Dry Chemical Surface Treatment Processes

#### 3.2.1. Plasma Treatment

Due to its ability to interact at the physical and chemical levels, plasma treatment is effective in surface treatment and modification of engineering fibres. Plasma treatment can produce uniform surface treatment in less time than other associated methods. The plasma process is also environmentally friendly, with no hazardous chemicals used during the process [102]. The required functional groups can be introduced on the fibre surface by selecting different plasma reaction gases like air, nitrogen, or oxygen [79].

In the research work done by Xie et al. [76], carbon fibres were treated using atmospheric pressure oxygen-helium plasma, as shown in Figure 12. The SEM and AFM analyses showed that the plasma treatment increased the fibre surface roughness, resulting in an increase of 21% in IFSS between CF and PI after 32 s of plasma treatment. XPS analysis showed that oxygen-containing functional groups on carbon fibre surfaces reached their maximum levels after 32 s of plasma treatment. A further increase in treatment time did not offer a higher level of oxidation. An increase in the oxygen-containing functional groups and surface topology contributed to fibre/matrix interfacial strength. In a similar study, Chang et al. [107] used plasma-treated CFs and 2 wt.% of CNTs to enhance the mechanical properties of CF-reinforced PA6 composite laminate. The flexural, impact, and tensile strength increased by nearly 20%, 24%, and 24%, respectively. This result proved that using treated CFs along with CNTs can increase the interlocking effect between fibre and matrix better than using only CNTs.

Zhang et al. [108] treated the poly(p-phenylene benzobisoxazole) (PBO) fibres using oxygen plasma treatment and examined its impact on the interfacial bonding of PBO fibre reinforced poly(phthalazinone ether sulfone ketone) (PPESK) composite laminates. The oxygen plasma treatment increased the ILSS by 32%. In another study, treating the CF surface using air plasma for 15 min improved the interfacial adhesion between CF and poly phthalazinone ether sulfone ketone (PPESK) resin by approximately 13% [109].

Poly aryl ether ketone (PAEK) sheets and CFs were treated using argon plasma before fabricating the compression-molded CF/PAEK composite laminates [110]. The argon plasma treatment removed the surface contaminants, increased the surface polarity of PAEK films, and formed micro dents on the CF surface, creating strong mechanical interlocking between the fibre and matrix. The argon plasma treatment increased the tensile strength of CF/PAEK laminate by approximately 10%.

Enciso et al. [111] used low-pressure plasma-treated flax and coconut fibres to fabricate flax/low-density polyethylene (LDPE) and coconut fibre/LDPE composite samples. Flax fibres with more cellulose content than coconut fibres exhibited better mechanical properties after the water immersion test. Low-pressure plasma treatment (LPP) improved interfacial bonding between the LDPE matrix and natural fibre reinforcements, enhancing interfacial bonding to create a strong barrier against moisture and water absorption.


*Limitations:*


Prolonged plasma treatment can cause fibre damage. A reduction in single-fibre tensile strength was reported, indicating the deterioration of the carbon fibre surface due to the formation of nano pits after plasma treatment [76]. Therefore, the plasma treatment time needs to be carefully controlled.

#### 3.2.2. Ozone Treatment

Several studies have shown that ozone treatment of fibres can remove surface impurities and introduce oxygen-rich functional groups, which enhance adhesion to the matrix [112]. Employing a hot molding process, Li et al. [80] used ozone surface-treated carbon fibres to reinforce polyimide (PI) thermoplastic resin. The specimens were subjected to tensile tests and tribological analysis. After the ozone surface treatment of CFs, the friction coefficient and specific wear rate decreased by 11% and 20%, respectively, whilst the tensile strength of the composite increased by 15%. The oxygen-containing functional groups on the fibre surface, which increased following the ozone treatment, play an essential role in increasing the surface energy and adhesion between fibre and matrix.

#### 3.2.3. Gamma Irradiation

The gamma irradiation treatment creates polymer chain scission (degradation) in the thermoplastic matrix, and these separated chains increase the concentration of carbonyl groups in the polymer matrix [113]. As a result, the polymer chains form a network structure of cross-linked polymer chains. Gamma irradiation should be conducted at a specific parameter with a suitable amount of fibre reinforcement to create a hybrid system with strong fibre-matrix interfacial bonding because overexposure to gamma rays can break down the polymer chains into smaller fragments, reducing their physical properties [114].

The effects of integrating irradiated polypropylene (PP) compatibilizer into the PP matrix on the interfacial adhesion between the carbon fibre and PP matrix were studied by Karsli et al. [115]. Gamma cell 220 type-ɣ irradiator was used to irradiate PP samples at room temperature in the presence of air at a fixed dose rate of 0.13 kGy/h. The irradiation dose was applied over a range of 5, 10, and 20 kGy. It has been found that gamma irradiation increased the ultimate tensile strength by up to 30%. The results revealed that irradiated PP compatibilizer, together with unsized CF, can improve the interfacial adhesion between the carbon fibres and matrix materials. Radiation-induced oxidation and the limited chain of PP were assumed to be the significant factor contributing to the increased interfacial bonding between PP and CF.

Hasioglu et al. [113] used extrusion and injection molding to prepare the neat, GF-reinforced, and CF-reinforced PC samples. The samples were gamma irradiated at four different doses of 10, 25, 50, and 75 kGy before being tested for their tensile properties. The gamma irradiation increased the tensile strength of PC, GF/PC, and CF/PC by 4%, 5%, and 2%, respectively. It was also found that GF/PC and CF/PC composites can be used till the total absorbed dose of 75 kGy gamma irradiation is reached. The higher gamma dose created higher chain scissions due to the formation of phenyl and phenoxy radicals, along with the reduction in the intensities of carbonyl and ether bonds in PC composites.

Similarly, powdered date pit/GF-reinforced linear low-density polyethylene (LLDPE) was fabricated using extrusion and injection molding, and the composites were treated using gamma irradiation [116]. The gamma irradiation increased the tensile and flexural strength of the LLDPE composite by approximately 100% and 525%. However, overexposure beyond 75 kGy caused the polymer crosslinking created by gamma absorption to break down and decrease the interfacial bonding.

After gamma irradiation treatment, a natural fibre-reinforced thermoplastic made of alfa fibre/PP composite was tested for its mechanical properties [117]. Gamma irradiation of PP composite increased the concentration of hydroperoxides, resulting in chain scission and subsequent physical property loss. No significant improvement in tensile strength was observed, but the storage modulus of irradiated samples increased by 18%.

Nitrile butadiene rubber (NBR) reinforced low-density polyethylene (LDPE) prepared using an extrusion process was irradiated using gamma rays and tested for its physio-chemical properties [114]. The gamma treatment made the pure polymer matrix brittle after crosslinking, but the addition of NBR to the LDPE matrix made the polymer composite stronger and more elastic. The nitrile groups in NBR also actively participated in the radiation crosslinking process, increasing the storage modulus to approximately 70%.

#### 3.2.4. Electron Beam Irradiation

The electron beam (EB) irradiation was used to activate the intermolecular interaction between the fibre and matrix at the interface region. Usually, EB irradiation was conducted on fibre-reinforced thermoplastic composite samples. Sometimes, fibre and matrix sheets were EB irradiated separately before being hot-pressed to form composite laminates. The EB irradiation provides two main effects to increase the load transfer and interfacial bonding between the fibre and matrix [118]. One is the cross-linking of the thermoplastic matrix for increased load transfer from matrix to fibre. The other effect is forming and bonding oxygen-based surface functional groups at the fibre-matrix interface. These two effects exhibit synergistic contributions to enhance the interfacial bonding and the subsequent mechanical properties of thermoplastic composites. It must be noted that conducting the EB irradiation on a thermoplastic composite using dosage parameters beyond the optimal parameters assigned for the material can gradually reduce the oxygen content on the fibre surface, producing a composite with poor physical properties [119].

Commercial long-fibre thermoplastic pellets made of CF and high-density polyethylene (HDPE) were mixed with HDPE pellets and injection molded to form CF/HDPE composite samples [119]. These composite samples were EB irradiated at different absorbed dose rates. For the optimal condition of 5 MeV beam energy, 400 kGy absorbed dose and 6.8 kGy/s absorbed dose rate the EB irradiation created potent hydroxyl (OH) intermolecular bond and oxygen content within the carbon fibre-reinforced thermoplastic (CFRTP) composite, which contributed to the increased interfacial interaction between CF and HDPE matrix. It was also observed that the tensile strength of the CF/HDPE composite increased with an increase in the absorbed dose of EB irradiation. Results showed that EB irradiation of 2000 kGy can increase the tensile strength of CF/HDPE laminate by 28% [118].

CF was electron beam irradiated under oxygen-rich nitrogen gas before hot pressing with PP sheets to form CF-reinforced PP composites [120]. The electron beam irradiation created strong covalent bonds between the carbon atoms present in CF and PP matrix. Performing EB irradiation in oxygen-rich conditions created oxygen-assisted intermolecular bonding between CF and PP matrix. The intermolecular chemical bonding increased the bending strength of treated composites by up to 26% over the untreated samples.

Kim et al. [121] used PU polymer sizing and EB irradiation to increase the CF and PA6 matrix interfacial bonding. CF coated with PU sizing agent was melt compounded with PA6 pellets and compression molded to form CF/PA6 composites. These composites were then EB irradiated to form cross-linking molecular chains in the PA6 matrix. The interaction between the urethane group in PU sizing and the amide group in PA6 also significantly improved the fibre-matrix interfacial bonding. The synergistic effects of polymer sizing and EB-irradiated PA6 cross-linking improved the storage modulus by up to 198% compared to untreated samples.

EB irradiation was also successfully utilized in forming cross-linkages in bio-based thermoplastic polymers. Szabó et al. [122] used EB irradiation on CF-reinforced cellulose propionate-derived thermoplastic composite samples to increase the fibre strength and polymer cross-linking within the composite. The EB treatment increased the tensile strength by approximately 29%.

### 3.3. Discussion on Fibre Surface Treatment

Fibre surface treatment achieved through wet-chemical methods, such as chemical coating, acid treatment, and polymer sizing, is widely used in industrial applications to fabricate thermoplastic matrix composite laminates with enhanced mechanical properties [75]. Wet chemical treatment can process a large number of fibres with minor fibre damage. The interfacial and interlaminar properties obtained from different fibre surface treatment methods are tabulated in Table 3. The advantages and disadvantages were presented in Table 4. Recent research shows that polymer colloids from PP can be used to size fibre surfaces and improve the interfacial strength between CF and different matrices such as PP [92] and PA6 [90,91,92]. Thermoplastics are poor in forming covalent chemical bonds with active chemical functional groups in either fibre surface or sizing. Thus, unless a compatible polymer sizing agent such as PP colloids to strengthen the CF/PP composite is used, the interfacial bonding will not be improved. In industrial applications, cost-effective processes, including acetone or intense acid treatment, are preferred to remove sizing agents and add surface roughness to the fibre material. Desizing the fibres and increasing their surface roughness are the essential factors to enhance the interfacial bonding between the fibre and thermoplastic matrix [123]. The surface roughness of fibre achieved after fibre treatment gives rise to interfacial strengthening factors, such as surface wettability, surface energy, and mechanical interlocking between fibre and matrix. The main disadvantage observed in all fibre treating methods is the damage caused to the fibres, which impairs their strength when the process is prolonged. This fibre damage can significantly worsen when reinforcing high-temperature thermoplastics like PEEK and PPS. The damage caused by the fibre treatment methods to the fibres should be carefully monitored while treating them to prevent or reduce its effect on the mechanical properties of composite laminates after fabrication.

## 4. Summary and Conclusions

Compared to thermoset matrix composites, the application of thermoplastic composites in high-performance engineering applications is limited. The main reasons are the weak fibre-matrix interfacial bonding, complex fabrication process, incompatibility while mixing with other thermoplastics, and the costly materials involved in the fabrication of thermoplastic composites. This review article discussed the results of fibre-matrix interfacial engineering techniques applicable to thermoplastic composites based on research findings, strengthening mechanisms, benefits, and limitations. The first part of the review article presented an overview of recent advances in integrating nanofillers at the fibre-matrix interface to strengthen the interfacial bond. The second part of the review article focused on fibre treatment methods involving wet chemicals and dry fibre treatment techniques to improve mechanical interlocking at the fibre-matrix interface.

In addition to variations in interfacial performance, cost, and safety hazards, a significant difference between the nanofiller inclusion and fibre treatment methods is the extent of fibre damage and the subsequent reduction in fibre strength. Nanofillers can be integrated at the fibre-matrix interface using varied methods that include EPD, CVD, spray coating, nano-sizing, nano-grafting, and flame synthesis. These nano-engineering methods have increased the fibre-matrix interfacial bonding without adversely impacting the fibre strength. By contrast, fibre strength was severely reduced when fibre treatment methods were used to improve fibre-matrix interfacial properties. The most significant fibre strength reduction was observed in the case of dry surface treatments.

Thermoplastic composites are generally manufactured using high-temperature processes such as hot-pressing. Applying heat to the fibre/matrix can adversely affect the fibre surface and reduce fibre strength, leading to poor load transfer between fibre and matrix. Therefore, methods that compromise fibre integrity prior to composite fabrication are not recommended. Integrating nanofillers at the fibre-matrix interface is preferred since this method can improve fibre-matrix interfacial bonding without damaging the fibres and reducing their strength.

Another challenge in selecting an interfacial engineering method for a thermoplastic composite revolves around the ability of the method to form a robust mechanical interlocking between fibre and thermoplastic matrix. Unlike thermosetting resins, thermoplastics are chemically inactive and cannot form covalent chemical bonds even with surface-functionalized nanoparticles or fibres. Therefore, interfacial bonding can only be enhanced by creating anchor points on the fibre surface, enabling mechanical interlocking between the fibre and the matrix. Including nanofillers at the fibre-matrix interface can increase the fibre surface roughness promoting fibre-matrix interlocking without adversely affecting the fibre strength.

While nanoparticles can hardly form chemical bonds with thermoplastic matrix, they can be grafted onto the fibres. The anchoring points created by fibre surface-grafted nanoparticles can provide improved interlocking between fibre and matrix in thermoplastic composites. Interfacial engineering techniques like nano-coating, flame synthesis of CNTs, and nanoparticle-spraying on fibre surfaces can be chosen to increase the fibre-matrix bonding via economic and efficient steps during the fabrication of thermoplastic composites.

## Figures and Tables

**Figure 1 polymers-15-00415-f001:**
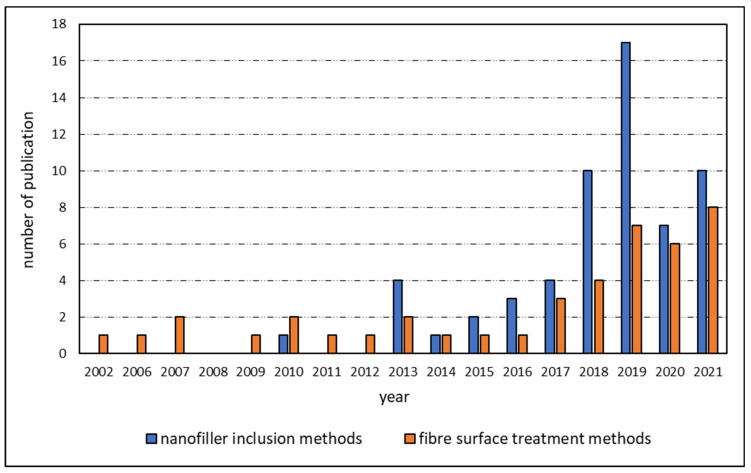
Survey of research papers related to interfacial engineering of thermoplastic composites.

**Figure 2 polymers-15-00415-f002:**
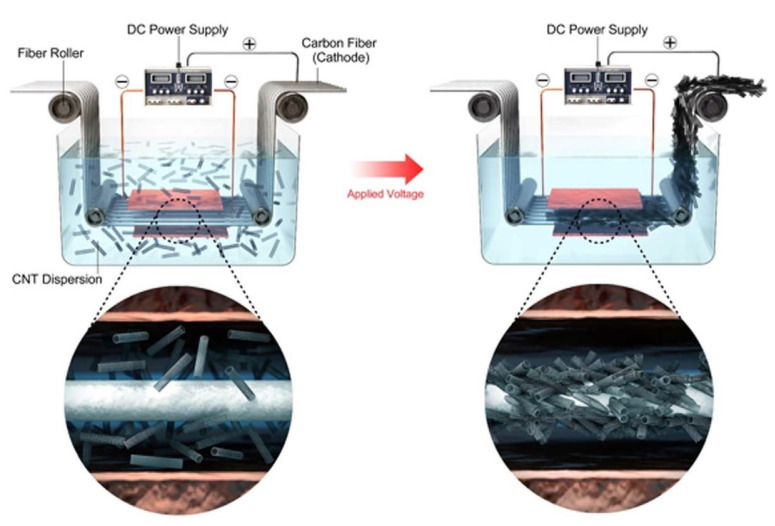
Experimental set-up for CNT deposition on CF using EPD (reprinted with permission) [13].

**Figure 3 polymers-15-00415-f003:**
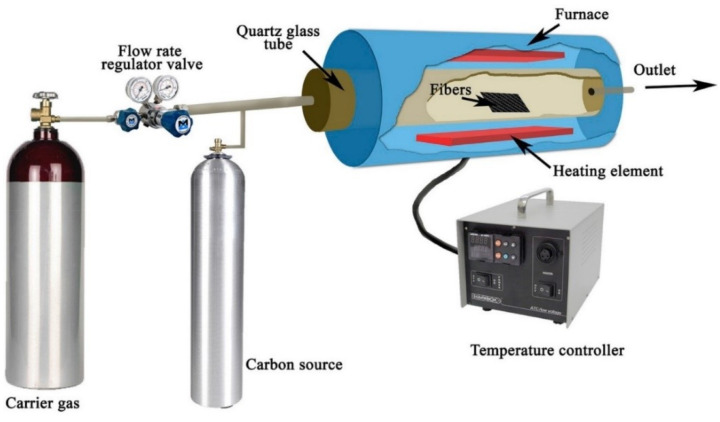
Experimental set-up of Chemical vapor deposition (reprinted with permission) [18].

**Figure 4 polymers-15-00415-f004:**
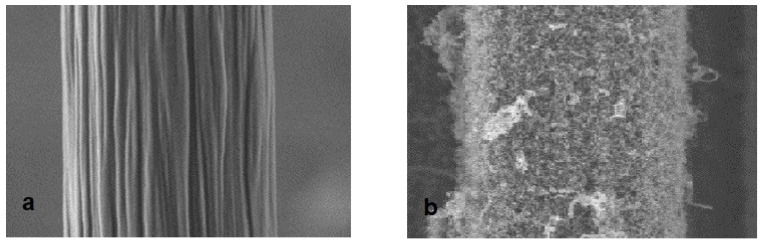
SEM image of 3 µm showing (**a**) acetone-treated CF, (**b**) CNT deposited CF via CVD [19].

**Figure 5 polymers-15-00415-f005:**
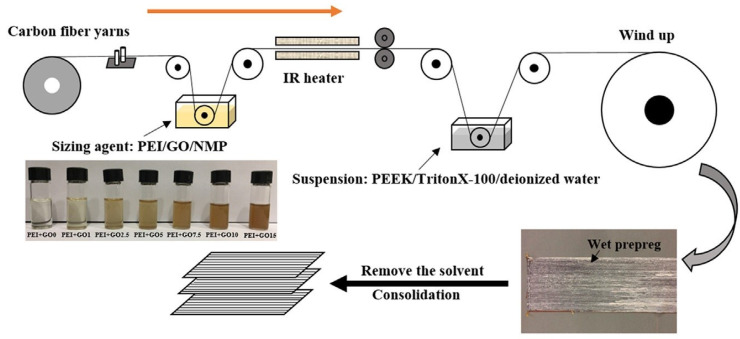
Experimental set-up of PEI+GO deposition over carbon fibres using direct immersion sizing method (reprinted with permission) [26].

**Figure 6 polymers-15-00415-f006:**
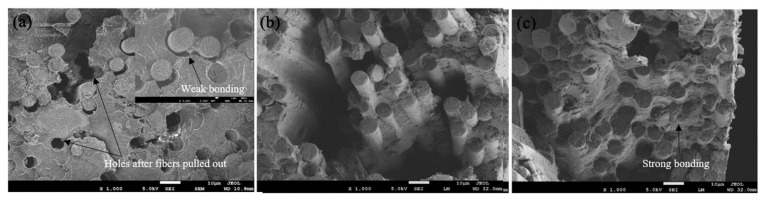
SEM images of flexural-tested CF/PEEK composites after fracture: (**a**) bare fibre without sizing, (**b**) PEI+GO1, (**c**) PEI+GO10 (reprinted with permission) [26].

**Figure 7 polymers-15-00415-f007:**
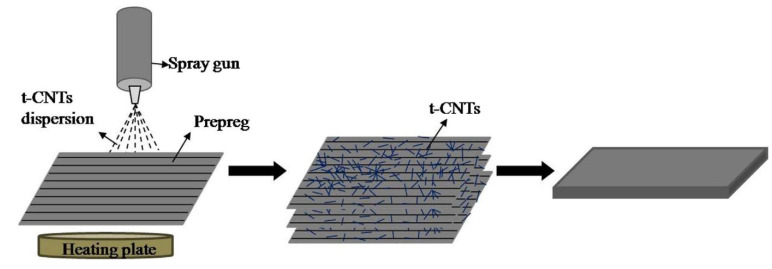
Preparation of CF/CNT/PEEK multiscale composite using spray gun technique (reprinted with permission) [51].

**Figure 8 polymers-15-00415-f008:**
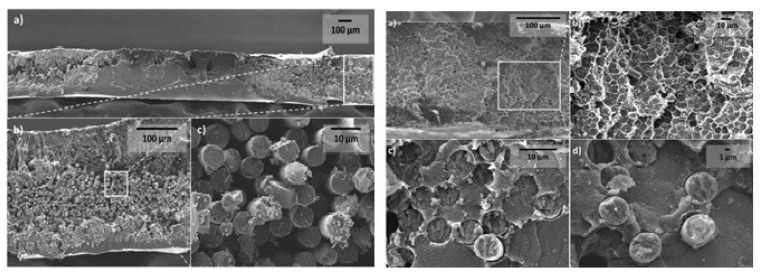
Fracture surfaces of AgNWs modified CF/PEKK laminates fabricated by film stacking method showing poor matrix impregnation with voids (**left**) and powder impregnation method showing improved matrix impregnation with no voids (**right**) at different magnifications (reprinted with permission) [52].

**Figure 9 polymers-15-00415-f009:**
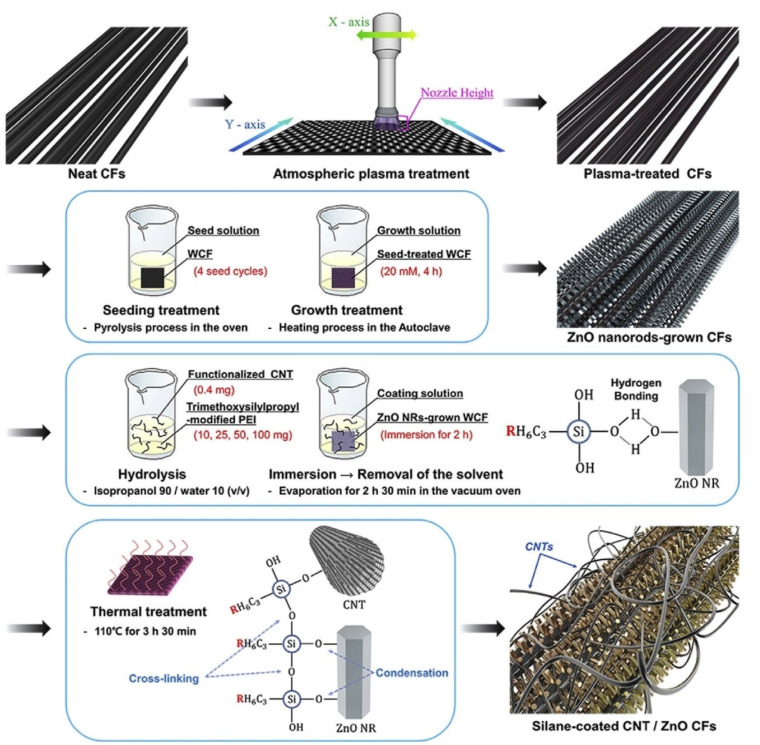
Processing schematics of ZnO NRs growth, CNTs-modified silane coating on CF surface (reprinted with permission) [54].

**Figure 10 polymers-15-00415-f010:**
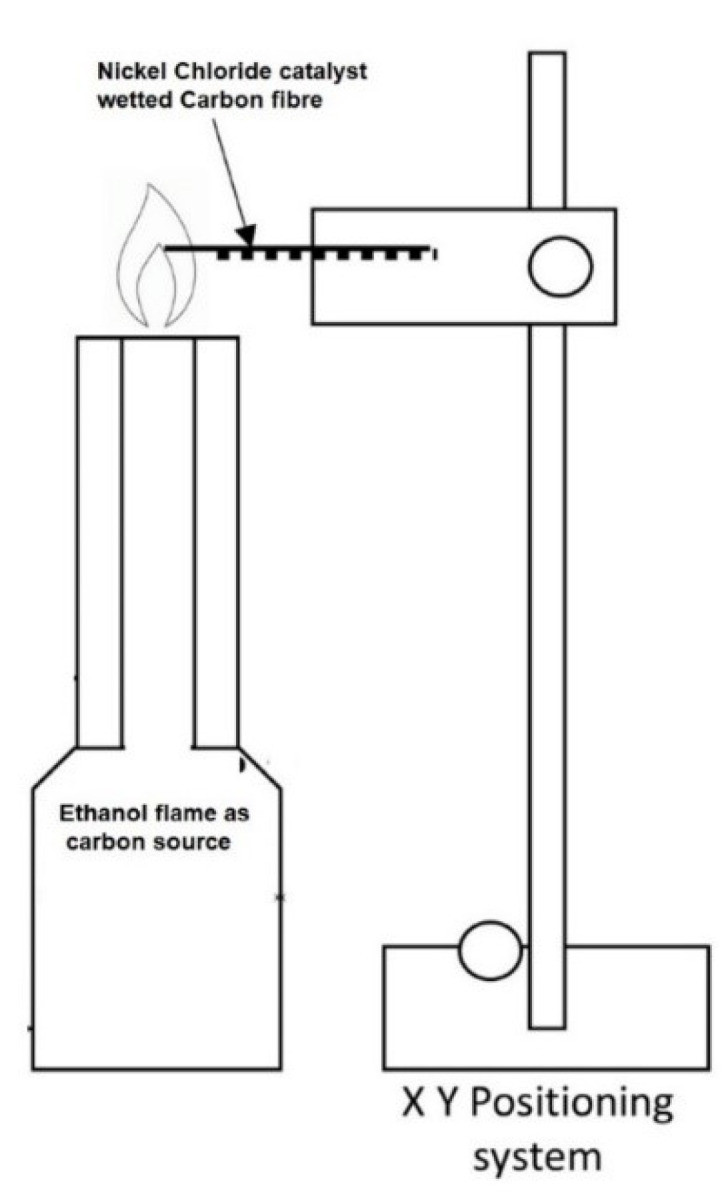
Experimental set-up to grow CNTs on carbon fibres through flame synthesis process (reprinted with permission) [59].

**Figure 11 polymers-15-00415-f011:**
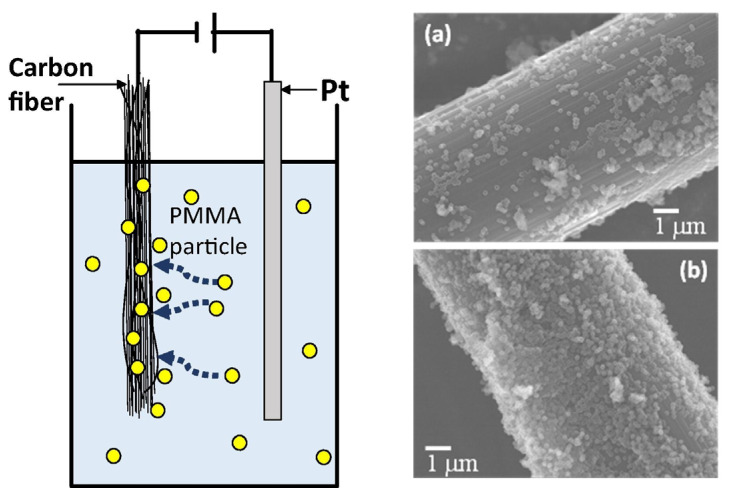
Depositing PMMA polymer colloid particles on CF via EPD (left) and SEM image of PMMA modified CFs at an applied voltage of (**a**) 6.5 V, (**b**) 20 V (right) (reprinted with permission) [90].

**Figure 12 polymers-15-00415-f012:**
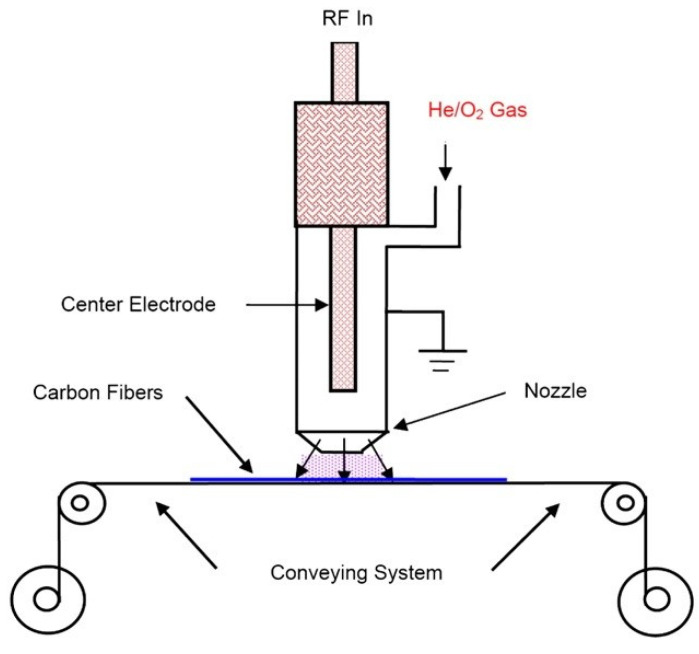
Plasma treatment on carbon fibres (reprinted with permission) [76].

**Table 1 polymers-15-00415-t001:** Testing results from various nanoparticle inclusion methods.

Nanoparticle Inclusion Method	Specimen	% Of Improvement in					Ref.
		IFSS	ILSS	ILFT	Tensile Strength	Tensile Modulus	
EPD	CF/MWCNT/PC	68%	-	-	-	-	[10]
	CF/CNT/PC	-	-	-	47%	58%	[14]
	MWCNT/PI	-	-	-	14%	50%	[9]
	CF/MWCNT/PPS	42%	-	-	-	-	[13]
	CF/CNT/PPEK	36%	-	-	-	-	[15]
CVD	CF/CNT/PP	35%	-	-	-	-	[18]
	CF/CNT/PP	35%	-	-	-	-	[18]
	CF/CNT/PA6	50%	-	-	-	-	[19]
	CF/CNT/PI	-	-	-	30%	-	[17]
Sizing with nanoparticles	CF/PEI+GO/PEEK	44%	12%	-	-	-	[26]
	GF/MWCNTs/PPS	-	23%	25% Mode I	-	-	[36]
	CF/nano-silica/PEEK	-	16%	-	-	-	[5]
	CF/PEI-ZIF 67/PEEK	40.5%	-	-	-	-	[48]
	CF/PEI+GO/PP	102%	-	-	48%	-	[24]
	CF/Fe_3_O_4_-RGO/PI	-	159%	-	-	-	[46]
	GF/MWCNT/EVA	49%	-	-	-	-	[44]
	CF/MWCNT/PEEK	-	55%	-	-	-	[47]
	CF/RGO/PA6	-	71%	-	-	-	[28]
	CF/s-PSF+GO/PEEK	-	128%	-	-	-	[29]
	CF/GO+Fe_3_O_4_/PP	-	32%	-	-	-	[33]
	CF/CNT/MPSR	-	42%	-	-	-	[38]
	CF/PI+CNT/PEEK	-	71%	-	-	-	[39]
	CF/HPEEK-g-MWCNT/PEEK	-	55%	-	-	-	[47]
	CF/PEI+ZIF-67/PEEK	41%	-	-	-	-	[48]
	CF/nano PEI/PVC	21%	-	-	-	-	[49]
	CF/nano PEI/PC	38%	-	-	-	-	
	CF/nano PEI/PA6	53%	-	-	-	-	
	CF/nano PEI/PP	50%	-	-	-	-	
	CF/nano PEI/PA66	43%	-	-	-	-	
	CF/nano PEI/PEI	58%	-	-	-	-	
	SCF/GO/PP	-	-	-	94%	-	[30]
	SCF/TPI-GO/PEI	-	-	-	13%	-	[31]
	LRGO+APP/TPU	-	-	-	190%	-	[32]
	GF/MWCNT/PPS	-	-	-	126%	-	[40]
	GF/PEI-CNT/PA6	-	-	-	7%	-	[41]
	Flax fibre/CNC/PP	13%	-	-	-	-	[50]
	Flax fibre/CNC+XG/PP	22%	-	-	-	-	
Spray gun	CF/CNT/PEEK	-	35.8%	-	-	-	[51]
	BF/SiO_2_+PCL/PLA	-	-	110% Mode I	29%	-	[53]
Chemically grafting	CF/MWCNTs/PA6	-	34%	-			[57]
	CF/Silica/MPSR	-	45.6%	-			[72]
	WCF/ZnO NR+CNT/PA6	-	-	-	41%	32%	[54]
Flame synthesis	CF/CNT/PEI	47%	-	-			[61]
	SGF/CNT/PLA	-	-	-	33%	43%	[62]
Nano modified monomer	CF/xGNP/PA6	-	-	-	18%	-	[63]
Nano inclusion in an extruder	GF/MWCNT/PP	-	-	-	76%	127%	[64]
	CF/CNT/PP	186%	-	-	7%	9%	[67]
	PA6 fibre/acid oxidized GNP	-	-	-	76%	70%	[68]

**Table 2 polymers-15-00415-t002:** Advantages and disadvantages of various nanoparticle inclusion methods.

Nanofiller Inclusion Method	Advantages	Disadvantages
Electrophoretic deposition (EPD)	Automated process suitable for large scale applications, it works well with electrically conductive fibre materials like carbon fibres.	The conducting current may damage fibres, non-uniform deposition and agglomeration of nanoparticles can occur, not suitable for electrically insulative fibres like glass and aramid.
Chemical vapor deposition (CVD)	Homogeneous thickness of nanoparticle deposition.	Expensive method suitable to process only small fibre surface area, catalyst contamination during processing and thermal damage to fibres can occur.
Direct immersion sizing (DIZ)	Simple and easy process suitable for processing large surface area of fibres.	Agglomeration and heterogeneous distribution of nanoparticles on fibre surface can occur.
Spray gun technology	Lesser void formation at the interfacial region, high matrix impregnation quality, nanoparticles can be deposited both at the surface and deep in between the fibres.	The coating thickness and uniform distribution of nanoparticles around the fibre surface is difficult to control.
Nanoparticle grafting	Chemical bond formation at the fibre matrix interface, high density of nanoparticles can be deposited on fibre surface.	Toxic chemicals are used, processing large fibre area and controlling the grafting density of nanoparticles is difficult.
Flame synthesis process	Longer and denser nanoparticles can be formed on fibre surface.	Toxic chemicals are heated while processing, Precise control of nanoparticle length is difficult. It is relatively new method which requires extensive optimization of processing parameters to synthesis CNTs on fibre surface.

**Table 3 polymers-15-00415-t003:** Testing results from various fibre surface treatment methods.

Fibre Surface Treatment Method	Specimen	% Of Improvement in				Ref.
		IFSS	ILSS	Tensile Strength	Tensile Modulus	
Heat treatment	CF/PEEK	15%	-	-	-	[77]
Fibre sizing	CF/PEKK/PEEK	-	70%	-	-	[87]
	CF/PEI/PEEK	17%	16%	-	-	[80]
	CF/SA-PI/PEEK	-	24%	-	-	[83]
	CF/PA/Elium	35%	-	-	-	[89]
	CF/PMMA colloid/PMMA	64%	-	-	-	[91]
	CF/PPS-COOH/PPS	28%	-	15%	-	[93]
	CF/PU/PC	-	25%	-	-	[88]
	CF/S-PEEK/PEEK	-	46%	11%	-	[81]
	GF/S-PEEK/PEEK	-	-	21%	-	[82]
	CF/silane + PPS/PPS	-	-	5%	-	[94]
	CF/PEI+PDA/PU	-	-	39%	59%	[95]
	CF/PA-SI/PEEK	41%	-	-	-	[48]
	CF/PP colloid/PP	300%	-	-	-	[92]
	CF/PP colloid/PA6	100%	-	-	-	[92]
	CF/An-biPEEK/PEEK	138%	67%	-	-	[84]
	CF/PEEK-NH_2_/PEEK	-	43%	-	-	[85]
Chemical grafting	CF/NH_2_-PASS/PASS	28%	72%	-	-	[99]
Chemical coating	Date palm leaf fibre/KOH+ Corn starch/PVA	-	-	61%	-	[101]
	Pineapple leaf fibre/alkaline/PA6	-	-	31%	8%	[100]
RES	GF/PTFE	-	-	21%	-	[105]
plasma treatment	PBO/PPESK	-	32%	-	-	[108]
	CF/PPESK	-	14%	-	-	[109]
	CF/PI	21%	-	-	-	[76]
	CF/CNT/PA6	-	-	24%	-	[107]
	CF/PAEK	-	-	10%	-	[110]
Ozone treatment	CF/PI	-	-	15%	-	[81]
Gamma irradiation	SCF/PP	-	-	30%	-	[115]
	PC	-	-	4%	-	[113]
	GF/PC	-	-	5%	-	
	CF/PC	-	-	2%	-	
	GF/powdered date pit/LLDPE	-	-	100%	-	[116]
Electron beam irradiation	CF/HDPE	-	-	28%	-	[118]
	CF/cellulose propionate derived thermoplastic	-	-	29%	-	[122]

**Table 4 polymers-15-00415-t004:** Advantages and disadvantages of various fibre surface treatment methods.

Fibre Surface Treatment Method	Advantages	Disadvantages
Wet-chemical methods Fibre sizing Chemical grafting Chemical coating Rare earth solution treatment	Direct chemical reaction resulting in a chemical bond formation between the functional groups of fibre surface and polymer matrix, large number of fibres can be processed, more suitable for industrial applications.	Toxic chemicals are involved in the chemical methods. Treating fibres using strong acids for prolonged duration can cause fibre damage.
Dry-mechanical interlocking methods Plasma treatment Ozone treatment Gamma irradiation Electron beam irradiation Heat treatment	Increases surface roughness, surface tension and wettability of fibre surface, introduces oxygen-based functional groups on fibre surface supporting its bonding with polymer matrix.	Damages the fibre strength while increasing the surface roughness, processing large area of fibres can be difficult, radiation and high temperature hazards are major health risks while using this process.

## Data Availability

No new data were created.

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
