# Peer review of "Interfacial Engineering Methods in Thermoplastic Composites: An Overview"

_polymers, 2023, doi:10.3390/polym15020415_

Round 1

Reviewer 1 Report

This paper investigated different interfacial enhancing methods used in thermoplastic composites.

The paper can be accepted after considering the following recommendations.

(1) When comparing the list of Nanofiller inclusion methods, it can be further subdivided, for example, the Electrophoretic deposition method can be listed in a separate table, on the one hand, the table is slightly larger, on the other hand, it can also add some other aspects of the comparison.

(2) Some methods have been introduced with their limitations elaborated, some have not, and this could be added if possible.

(3) Please avoid lumped citations, such as [2-4]

(4) The writing needs to be further polished. The current version still has grammar mistakes and typo errors.

(5) Some of the figures come from existing literature. The authors are advised to ask for permission from the publishers to use these figures.

Reviewer 2 Report

The paper entitled “Interfacial Engineering Methods in Thermoplastic Composites: An Overview” summarized the interfacial enhancing methods used in thermoplastic composites. I think this topic is interesting, and the authors have well summarized different methods. But before publishing in this journal, I suggest the authors address the following comments.

1.      There are many language mistakes in the manuscript. For example, first sentence in the abstract part “ The paper critically analyses different interfacial enhancing methods used in thermoplastic composites”. analyses should be analyzed. Please revise these mistakes carefully.

2.      Please check the format of the references. Many references lost the page number.

3.      Please update the reference, I think the latest two years' references are missing the in this manuscript.

4.      It is better to draw a table to summarize the advantages and disadvantages of different methods to treat the fibers.

5.      The manuscript is mainly focused on the surface treatment of fibers. I suggest the authors revise the title to make it more concise and clear.

Round 2

Reviewer 2 Report

The authors have addressed my comments. I recommend it to be published in this journal.